# Assessment of the Potential of a Native Non-Aflatoxigenic *Aspergillus flavus* Isolate to Reduce Aflatoxin Contamination in Dairy Feed

**DOI:** 10.3390/toxins14070437

**Published:** 2022-06-27

**Authors:** Erika Janet Rangel-Muñoz, Arturo Gerardo Valdivia-Flores, Sanjuana Hernández-Delgado, Carlos Cruz-Vázquez, María Carolina de-Luna-López, Teódulo Quezada-Tristán, Raúl Ortiz-Martínez, Netzahualcóyotl Mayek-Pérez

**Affiliations:** 1Centro de Ciencias Agropecuarias, Universidad Autónoma de Aguascalientes, Aguascalientes 20100, Mexico; janet.rangel@edu.uaa.mx (E.J.R.-M.); carolina.deluna@edu.uaa.mx (M.C.d.-L.-L.); teodulo.quezada@edu.uaa.mx (T.Q.-T.); raul.ortiz@edu.uaa.mx (R.O.-M.); 2Instituto Politécnico Nacional, Centro de Biotecnología Genómica, Reynosa 88710, Mexico; shernandezd@ipn.mx; 3Instituto Tecnológico de México, Aguascalientes 20330, Mexico; cruva18@yahoo.com.mx; 4Unidad Académica Multidisciplinaria Reynosa–Rodhe, Universidad Autónoma de Tamaulipas, Reynosa 88779, Mexico; nmayeklp@yahoo.com.mx

**Keywords:** mycotoxins, biological control, aflatoxin biocontrol agents, Mexico, dairy cows

## Abstract

*Aspergillus* species can produce aflatoxins (AFs), which can severely affect human and animal health. The objective was to evaluate the efficacy of reducing AF contamination of a non-aflatoxigenic isolate of *A. flavus* experimentally coinoculated with different aflatoxigenic strains in whole plant (WP), corn silage (CS), immature grains (IG) and in culture media (CM). An L-morphotype of *A. flavus* (CS1) was obtained from CS in a dairy farm located in the Mexican Highland Plateau; The CS1 failed to amplify the AFs biosynthetic pathway regulatory gene (aflR). Monosporic CS1 isolates were coinoculated in WP, CS, IG and CM, together with *A. flavus* strains with known aflatoxigenic capacity (originating from Cuautitlán and Tamaulipas, Mexico), and native isolates from concentrate feed (CF1, CF2 and CF3) and CS (CS2, CS3). AF production was evaluated by HPLC and fungal growth rate was measured on culture media. The positive control strains and those isolated from CF produced a large average amount of AFs (15,622 ± 3952 and 12,189 ± 3311 µg/kg), whereas *A. flavus* strains obtained from CS produced a lower AF concentration (126 ± 25.9 µg/kg). CS1 was efficient (*p* < 0.01) in decreasing AF concentrations when coinoculated together with CF, CS and aflatoxigenic positive control strains (71.6–88.7, 51.0–51.1 and 63.1–71.5%) on WP, CS, IG and CM substrates (73.9–78.2, 65.1–73.7, 63.8–68.4 and 57.4–67.6%). The results suggest that the non-aflatoxigenic isolate can be an effective tool to reduce AF contamination in feed and to minimize the presence of its metabolites in raw milk and dairy products intended for human nutrition.

## 1. Introduction

*Aspergillus* spp. is a genus of filamentous, saprophytic fungi that is a natural inhabitant of soil and agricultural products, especially grains and seeds. Several species of *Aspergillus* can produce aflatoxins (AFs), which can contaminate crops and remain in food products, so these compounds can affect both public health, agriculture, livestock, and agribusiness [1]. AF contamination occurs in the agricultural field before, during or after harvest as well as in the transport, storage or processing of agricultural products, thus affecting entire food chains [2].

AFs are secondary metabolites of the fungus and possess carcinogenic, immunosuppressive, nephrotoxic and hepatotoxic properties, thus representing a serious health risk to humans and animals. The original forms of AFs (AFB_1_, AFB_2_, AFG_1_ and AFG_2_) are neither toxic nor mutagenic, but when ingested in food or feed they are distributed within the body tissues, where a biotransformation process occurs by the mixed-function oxidase system (cytochrome P_450_), modifying their chemical form as an AF-8,9-epoxide. The epoxides are highly reactive, exert an electrophilically attack on subcellular structures and cause damage to nucleic acids [3]. Detoxification processes (i.e., hydrolysis, demethylation or ketoreduction) after biotransformation form soluble and less toxic products (AFM_1_, AFM_2_, AFQ_1_, AFQ_2_, AFP_1_, AFP_2_, Aflatoxicol, AF-FAPy and AF-N^7^guanine), which are eliminated by excretions and milk [4].

Because of the health risk caused by AF contamination, in several countries and geographical regions maximum permissible limits (MPLs) have been imposed on the tolerable concentration of AFs in foods and feed. To maintain food safety below the MPL and avoid the undesirable effects of AFs, various physical and chemical control methods have also been developed to process feed ingredients, decrease the growth of fungal microflora and reduce the content of AFs [5]. The major methods have been focused on good agricultural practices (crop rotation, strategic irrigation, insect control, fungus-resistant plants) and optimal storage conditions (ambient temperature, relative humidity and time of consumption) can be complemented with physical grain-processing techniques (sieving and extrusion) and fungal microbiota inhibitors (benzoic, acetic, sorbic and propionic acids, ozone gas, etc.) [2,4,6,7]. In addition, the toxicity of AFs has been decreased by using compounds that reduce their gastrointestinal absorption; the most used mineral sequestrants are some phyllosilicates, such as hydrated calcium and sodium aluminosilicates, bentonite and tectosilicates or zeolites; as well as organic sequestrants derived from yeasts and other microorganisms (mannoproteins and β-D-glucomannan from *Saccharomyces cerevisiae*, peptidoglycans from lactic-acid bacteria) [6,8,9]. In addition, some dietary supplements (curcumin, vitamins, Selenium, N-acetylcysteine, methionine, etc.) have been developed that prevent or reduce the oxidative stress induced by Afs; thus, they can decrease the toxic effects in animals [10,11]. All these methods of control of AFs in food have a wide range of efficacy, but none of them eliminate the risk of intoxication at high concentrations of AFs, nor the negative effects of ingestion of low doses for a long period; moreover, they generate an extra expense for farmers and do not prevent the abundance of toxigenic fungi in agricultural soils [3,4].

Therefore, the use of biological control strategies for the control of AF contamination by bacteria, yeasts and fungi has been of great interest [12,13]. It has been established that some fungal strains considered non-aflatoxigenic have a very limited capacity to produce these toxins and have the ability to interact with toxigenic strains growing within the same substrate and induce the reduction of AFs biosynthesis [14,15,16]. Non-aflatoxigenic strains of *A. flavus* have been associated with L-morphotypes, characterized by the development of large sclerotia (>400 µm) [17,18]. The lack of aflatoxigenic capacity of this type of fungi has also been related to the absence of genotypic information involved in the AFs biosynthetic pathway, especially the pathway regulatory gene (aflR) and the aflO and aflQ structural genes [19,20]. The mechanism underlying interactions between fungal strains that share substrates where they coexist has not been fully elucidated; some of the possibilities that have been proposed are the competitive inhibition of toxigenic strains, the thigmoregulation or mechanotransduction of surface physical signals through cytoskeleton filaments [21,22,23], as well as the chemical detection of organic compounds produced by strains of *A. flavus* strains [24]. All these mechanisms have the potential to regulate and reduce AF levels in toxigenic strains, thereby affecting their growth or the expression of genes involved in AF biosynthesis, and therefore, their accumulation in food.

As in other food chains, a mixture of *A. flavus* strains with and without aflatoxigenic capacity has been shown to occur in bovine milk production in the Mexican Highland Plateau [25,26]. This mixture of strains has occurred in grain-based concentrate feed and corn silage, as well as in the totally mixed ration of dairy cows, which is directly related to AFM_1_ contamination in raw milk. However, the efficacy and sustainability of non-aflatoxigenic strains of *A. flavus* for the biological control of AFs must be tested before they are released into the environment [2,13,18]. Therefore, the objective was to evaluate the reduction in AF contamination in whole plant (WP), corn silage (CS), immature grains (IG) and in culture media (CM) when a non-aflatoxigenic isolate of *A. flavus* was experimentally coinoculated with different aflatoxigenic strains.

## 2. Results

The non-aflatoxigenic *A. flavus* CS1 strain reduced the concentration of AFs in WP, CS, IG and in CM. This effect was demonstrated when CS1 was coinoculated together with control and native aflatoxigenic strains. The Cuautitlán and Tamaulipas strains used as the positive control produced a large average amount of AFs (15,622 ± 3952 µg/kg) on all substrates (Figure 1); however, when the positive control and CS1 strains were inoculated together, they reduced (*p* < 0.01) the average amount of AFs produced on all substrates (5052 ± 1920 µg/kg) compared to the concentration of AFs produced in the single inoculation. The efficiency shown by CS1 in reducing the amount of AFs produced on the four substrates by the Cuautitlán and Tamaulipas strains was 63.1% and 71.5%, respectively (Table 1).

Similarly, *A. flavus* strains obtained from concentrate feed (CF1, CF2 and CF3) for dairy cows produced on average a large amount of AFs (12,189 ± 3311 µg/kg) on all substrates (Figure 2); however, when inoculated together with the CS1 strain, the average amount of AFs produced on all substrates was decreased (*p* < 0.01) (1647 ± 502 µg/kg) compared to the concentration of AFs produced in the single inoculation. The efficacy shown by the CS1 strain in reducing the AFs amount by CF1, CF2 and CF3 strains on all substrates was 81.0 ± 3.3% (Table 1). On the other hand, *A. flavus* strains obtained from corn silage (CS1, CS2 and CS3) produced AFs at a lower concentration (126 ± 25.9 µg/kg) than the other aflatoxigenic isolates on all substrates (Figure 3); furthermore, when inoculated together with the CS1 strain, the AF amount was significantly decreased (*p* < 0.01) (26.7 ± 3.2 µg/kg) compared to the AF concentration in the single inoculation on all substrates. The efficacy shown by CS1 in reducing the amount of AFs produced by CS2 and CS3 strains on the four substrates averaged 51.1 ± 1.6% (Table 1).

The lowest AF concentration was detected when aflatoxigenic strains were inoculated on the WP (167 ± 19.1 µg/kg), whereas 10 times more was accumulated in CS (1721 ± 368 µg/kg). However, the AF concentration reached the maximum value when all strains had the conditions of IC kernels and CM (10,304 ± 1786 and 10,440 ± 1685 µg/kg, respectively). The average efficacy of the CS1 strain to reduce the amount of AFs produced by all toxigenic strains of *A. flavus* showed significant differences (*p* < 0.05) between WP and CS (Table 1), whereas this indicator showed an intermediate position in IG and CM. No differences (*p* > 0.05) were observed in the speed or surface area of mycelial growth in the different CM (25.4 ± 4.1 cm^2^/7 days) during the coinoculation of the toxigenic strains against the native CS1 strain of *A. flavus* (Figure 4).

The native strain of *A. flavus* CS1 was considered non-aflatoxigenic (Table 2) because it did not produce AFs and did not express the aflR gene (Figure 5); however, in all experimental isolates and positive control isolates (Cuautitlán and Tamaulipas) expression of this regulatory gene of the AFs biosynthetic pathway was observed. All isolates were identified as *A. flavus* by analysis of gene expression of an ITS region and CaM (Table 3). The nucleotide sequences obtained showed a high percentage of homology (90–99%) with other nucleotide sequences registered for *A. flavus* in public databases (Figure 6).

## 3. Discussion

When AFs produced by *A. flavus* contaminate crops, they reduce the nutritional properties of crops and products of agroindustrial interest. Modern biotechnological research tries to combine the reduction of these contaminants, the study of the biology of the fungus and the analysis of the toxic effects of AFs, so this research approach is essential and should be sustained as a long-term strategy, for both integrated crop management and storage and the use of chemical and physical methods to prevent the incidence and damage by aflatoxigenic fungi [5,27,28].

Biological control agents have been selected based on their efficiency in reducing AF production in toxigenic strains. In this work, CS1, the non-aflatoxigenic strain of *A. flavus*, reduced AF production both in the WP, CS, IG and CM; such efficiency in reducing AF production ranged from 46.7–92.9%, depending on the aflatoxigenic strain and evaluation substrate. Different authors have identified non-aflatoxigenic strains of *A. flavus* with the ability to reduce AF contamination in several cultivated plant species. For example, in Texas, USA, they reported [29] an 85–88% reduction in AF production in cultivated maize when applying a biopesticide called ‘Afla-Guard’, a non-aflatoxigenic isolate of *A. flavus*. On the other hand, researchers in Argentina [30] evaluated the competitive ability of *A. flavus* isolates that did not produce cyclopiazonic acid (associated with the non-aflatoxigenic condition of the strains), in coinoculations in maize grains with aflatoxigenic strains. These isolates reduced AF contamination in corn kernels by 6–60%. In that study, *A. flavus* strain ARG5/30 was identified as a candidate for development as a potential biocontrol agent in maize. Meanwhile, in China [31], researchers identified two non-aflatoxigenic isolates of *A. flavus* that reduced aflatoxin B_1_ and aflatoxin B_2_ contamination by 26–99% in groundnut grown under varying soil-moisture conditions.

Moreover, in agroecological zones in Ghana [32], non-aflatoxigenic strains of *A. flavus*, with superior competitive potential and wide adaptation, were selected and tested for efficacy as AF biological control agents. In the laboratory, AF biosynthesis was reduced from 87–98% compared to grains inoculated with the AF-producing strain alone, whereas under field conditions (100 crops, 50 with maize and 50 with groundnut), AF levels in treated isolates were lower than in untreated crops (70–100% in groundnut and 50–100% in maize). Therefore, it is proposed that the combined use of appropriate and adapted non-aflatoxigenic *A. flavus* isolates can displace AF-producing *A. flavus* populations, thus limiting AF contamination. 

In this study, a decrease in AF production by toxigenic *A. flavus* strains was observed, but no competitive inhibition effect of non-aflatoxigenic *A. flavus* strains was detected in the CM, because mycelial growth speed and surface area were not significantly different. However, it has been shown that non-aflatoxigenic strains of *A. flavus* can reduce aflatoxigenic fungal populations in soil [33,34]. Therefore, it is proposed that the selection of strains should be based on criteria such as their ability to colonize soils and grains after their release in crop fields.

In this work, lower AF production was observed in aflatoxigenic strains inoculated in WP, while in IC, CM and CS they accumulated to a greater extent, possibly because of the combination of better humidity and temperature conditions for their production. In contrast, it has been reported [31] that AF contamination in peanuts grown in China was higher in plants grown under water deficiency (84–99%) than under irrigated conditions (26 to 99%). Since the efficacy in controlling AF production in this study depended on the substrate and test conditions, it will be necessary for the CS1 strain to confirm its ability to biologically control AF production under natural growing conditions and corn silage production in Mexico. 

In our study, given the efficacy of the CS1 strain in different stages of cattle feed production, such as in field cultivation, in IG or in fermented silage, it is proposed as a strain for biological control of AFs in the production process of feed destined for dairy cow diets in central Mexico, considering that the use of non-aflatoxigenic isolates of *A. flavus* in biological control strategies of aflatoxigenic fungi offers the potential to improve food security, productivity and income opportunities for farmers, large and small, of susceptible crops such as corn. 

The use of non-aflatoxigenic *A. flavus* strains is not specifically designed to address the reduction of other mycotoxins that are also harmful to humans and animals when ingested [22]; moreover, some reports indicate that non-aflatoxigenic *Aspergillus* strains produce other mycotoxins in addition to AFs [34,35,36]. On the other hand, the existence of the sexual state of *Aspergillus* suggests the possibility that a non-aflatoxigenic strain can become toxigenic through sexual reproduction [37]. 

Some authors [13,18,27] suggest that when using non-aflatoxigenic strains as a biological control measure, some relevant circumstances should be considered, such as the genetic and reproductive stability of non-aflatoxigenic populations and the persistence of strains in field, depending on ecological and environmental conditions, as well as the mechanisms that allow maintaining control and enhancing the biological effectiveness of non-aflatoxigenic strains. On the other hand, it has been pointed out [27,38,39] that the selection of non-aflatoxigenic strains for the effective biocontrol of *A. flavus* should be carried out based on their adaptation to the environment, the cultivated plant species, and the type of soil where the biocontrol agent will be introduced. It is also important that biocontrol strategies be flexible and adaptable to constant changes in nutrients and microbiome populations and to climate change.

In this study, the non-aflatoxigenic *A. flavus* strain CS1 was shown to have the potential to reduce AF production (Table 1), and thus may be an effective tool to reduce AF contamination in cow feed and milk, as well as offering the possibility to improve food security, farm productivity and income opportunities for farmers of susceptible crops such as corn. However, before its practical use, it will be necessary to confirm its efficacy to control the production of AFs under natural conditions of forage production, during multiple years and diverse agroecological zones of central Mexico. Therefore, field evaluation should be based on criteria such as their ability to colonize soils and crops under specific ecological conditions, persistence in the field and genetic stability [2,13,18]. Finally, a phase that goes beyond the aims of this work is the design, testing and validation of industrial processes to produce the active ingredient fungi and the biocontrol product per se, as well as the preparation of dossiers for registration as biocontrol product, with product ecotoxicological data, instructions for use and commercialization according to national regulations.

## 4. Conclusions

This study identified a non-aflatoxigenic strain of *A. flavus*, called CS1, with the potential to reduce 46.7–92.9% of AF production in different forage corn products (WP, CS, IG) and in CM. Therefore, the non-aflatoxigenic CS1 isolate can be an effective tool to mitigate AF contamination in feed and milk, which could provide health, commercial and economic benefits to Mexican dairies.

Effective biocontrol by the CS1 strain of A. flavus should be validated according to its range of environmental adaptation and the crop and type of soil where it could be introduced, considering the relevant aspects on food safety and quality, as well as the pertinent Mexican regulations.

## 5. Materials and Methods

### 5.1. Fungal Strains

Six isolates from concentrate feed (CF1, CF2 and CF3) and corn silage (CS1, CS2, CS3) used in this study were obtained from a dairy-production unit located in the Mexican Highland Plateau (21°48′ N, 102°03′ W; 1986–2008 masl) where the samples (*n* = 267) of feed ingredients were collected for 24 months [25]. The CS1 isolate showed macroscopic and microscopic morphological characteristics coincident with the L-morphotype of *A. flavus* and was molecularly identified by amplification of an internal transcribed spacer (ITS1-5.8S-ITS2 RNAr) and calmodulin gene, but CS1 strain showed a negative result to amplification of the AFs biosynthetic pathway regulator gene, aflR, with absence of AF production (Table 2). Two strains of *A. flavus* (originating from Cuautitlán, State of Mexico and Río Bravo, Tamaulipas) with known aflatoxigenic capacity [7] and one non-aflatoxigenic strain [40] (AF36, NRRL 18543) were also included in the study design.

### 5.2. Treatment Design 

The evaluation of AF contamination produced by aflatoxigenic strains that interacted with the CS1 strain (Table 1) was carried out with five replicates for each treatment, in each of the following four circumstances (Figure 5): (A) direct inoculation in corn ears (in situ) under greenhouse conditions; (B) microsilage of the crushed corn plant, with controlled compression and air vacuum; (C) Spores inoculation in immature corn kernels in the laboratory, under controlled temperature and humidity conditions; and (D) seeding in CM in the laboratory. Spore production and application of the above isolates were obtained according to a previously reported mycological method [7] consisting of seeding each strain in monosporic cultures, incubated in darkness (10 days, 27 °C). Spores were suspended in a sterile solution of Tween 80 (0.1%) and quantified and diluted for inoculation (5 mL of inoculum: 7 × 10^7^ spores/mL) with paraffinic oil (1.0%) as the adherent medium. In each substrate, a negative control or sham was included, consisting of the application of the complete inoculation procedure but depositing only the inoculum diluent without spores.

### 5.3. In Situ Inoculation

In a greenhouse at the Centro de Ciencias Agropecuarias of the Universidad Autónoma de Aguascalientes, which was previously subjected to a thorough washing and disinfection process (sodium hypochlorite 3.0%), intermediate growth-cycle hybrid maize (P3055W, Pionner) shown to be susceptible to colonization by toxigenic *A. flavus* [7] was used. Kernels were disinfected with sodium hypochlorite, and germinated kernels were sown in pots with a substrate free of specific microorganisms (18 L). The experimental planting was carried out (27 May 2020) by placing germinated plants to obtain a density equivalent to 10,000 plants per ha. The average temperature in the whole agriculture period was 20.5 ± 5.2 °C. Irrigation was applied three times per week, completing a total irrigation lamina equivalent to 700 mm of water in the whole period. At planting, fertilization was carried out with a dose equivalent to 100-100-00 N P K plus 20 kg/ha of minor elements, while an additional amount of N (180 kg/ha) was applied weekly in the irrigation water from day 35 postplanting. At 95 days after germination, the spore suspension was instilled on the stigmas of each corn with the aid of a needle, and each corn ear was isolated with a hermetically sealed paper bag, according to a previously described method [41]. 

### 5.4. Immature Kernels

Intact ears were harvested 100 days after sowing, when the milk line of the kernels had reached approximately 75% distance from the kernel edge [7]. Kernels were separated from dehydrated corn, sterilized (121 °C, 15 min) and kept in airtight glass containers with lids (400 g/container). Inoculation of the kernels with spores of the CS1 strain and aflatoxigenic strains was performed using a sterile, noninvasive technique, and the humidity was adjusted to 15% by the addition of sterile distilled water. The flasks were shaken daily to prevent adhesion. 

### 5.5. Corn Silage

Following a previously informed method [42], a miniature corn silage model, or microsilage, was developed to evaluate AF contamination during ensiling of corn plants. All aerial parts of the plants were collected from 15 cm above the ground, the corn ears were removed, and the mean existing grain concentration (265 g/kg) was estimated. Crushing was performed using an electric hammer mill until a uniform particle size (2–3 cm) was achieved. The crushed aerial part was homogenized, and from the resulting pool, the samples were obtained. Each microsilage bag (1.5 kg) received IG (400 g) inoculated immediately beforehand according to the treatment scheme. The crushed forage was homogenized, deposited in double plastic bags and uniformly compacted (10 kg/cm^2^); a pneumatic vacuum pump was used to extract the air from the bag, which was then hermetically sealed. The microsilage were properly stored until the anaerobic fermentation process of the silage was completed (50 days). 

### 5.6. In Vitro Culture 

Using a fungal culture technique [43,44] from monosporic mycelia with 24 h of growth, each pair of facing isolates was placed equidistant (1.5 cm) from the center of the Petri dish (diameter: 9.0 cm) and incubated in darkness (28 °C, seven days) on potato dextrose agar, Czapeck, Rose Bengal and coconut agar (Figure 4). The area filled by mycelia was observed using a stereo microscope, and the surface area and growth rate of each strain were estimated using software (Leica MZ6/DFC320 G, Qwin Pro-Image Analysis system, Microsystems, Heerbrugg, Switzerland). At the end of the period, the fungal mass was obtained, and the AFs produced were quantified.

### 5.7. Quantification of Aflatoxins 

AF quantification was performed with the 990.33 AOAC official method [45]; the samples were processed using the solid-phase columns (SupelcleanTM LC-18 SPE tube, Sigma-Aldrich, St. Louis, MO, USA), methanol:water, acetic acid, tetrahydrofuran (THF) and hexane. The eluate was obtained with methylene chloride:THF; evaporated using a nitrogen stream; derivatized using trifluoroacetic acid; and injected in triplicate into the HPLC system with fluorescence detector (Varian Pro Star binary pump; FP detector 2020, Varian Associates Inc., Victoria, Australia), C18 column and column guard (LC-18 and LC-18; Thermo Fisher Scientific, Waltham, MA, USA). The concentration of AFs in the samples was estimated with software (Galaxie Ver. 1.9.302.530) and compared with the calibration curves of purified Afs (B1, B2, G1 and G2; St. Louis, MO, USA). The limit of quantification of the HPLC method for AFs was 2.5 µg/kg. The AF concentration was estimated based on the dry matter present in the WP, CS, IG, and CM. The amount of AFs produced during the ensiling process was estimated after corroborating the amount of AFs present in the grain that was added to the shredded plant material at the beginning of fermentation.

### 5.8. Molecular Analysis

DNA extraction, amplification and sequencing of monosporic cultures of each of the *A. flavus* isolates were carried out by standard methods [46,47,48,49]. The DNA obtained was compared against phage λ DNA concentrations (Thermo Fisher Scientific, Waltham, MA, USA) and visualized in a photodocumenter (Bio-Rad Molecular Imaging-Gel Doctm XR, Hercules, CA, USA). Amplification was performed with Go-Taq polymerase enzyme (Promega, Madison, WI, USA) and a thermal cycler (model 9700 Applied Biosystems). Fragments of the internal transcribed spacer region (ITS), the calmodulin gene (CaM) and the AF biosynthetic pathway regulator gene (aflR) were amplified using the primers ITS1, ITS4, CMDA7-F, CMDA8,R, aflR-F and aflR-R (5′-TCCGTAGGTGAACCTGCGG-3′; 5′-TCCTCCGCTTATTGATATATG-3′; 5′-GCCAAAATCTCTTCATCCGTAG-3′; 5′-ATTTCGTTCAGAATGCCAGGCAGG-3′; 5′-GGGATAGCTGTACGAGTTGTGCCAG-3′; 5′ TGGKGCCGACTCGAGGAAYGGGT-3′) (Eurofins Genomics, Lousville, KY, USA), respectively.

The PCR products obtained were separated by agarose-gel electrophoresis and observed with SYBR^®^ Gold and Orange DNA Loading Dye reagents (Thermo Fisher Scientific, Waltham, MA, USA) with molecular-weight-marker ladders (Axygen Biosciences, Union City, CA, USA); the resulting bands were observed in an imaging photodocumenter (BIO-RAD Molecular Imaging^®^ GEL DOCTM XR, Hercules, CA, USA) with the Quantity One software. PCR products were purified with ExoSAP-IT PCR Product Cleanup reagent (Afflymetrix, Thermo Fisher Scientific Inc., Waltham, CA, USA), which were sequenced into forward (F) and reverse (R) strands with the dideoxy method (44). The samples were injected into a sequencer (ABI 3730XL Genetic Analyzers) and the resulting nucleotide sequences were recorded in electropherograms, which were visualized with the Chromas Lite software and the identity was compared using the Basic Local Alignment Search Tool (BLAST), with other nucleotide sequences of *A. flavus* in the records of the National Center for Biotechnology Information (NCBI) (Table 3). Finally, the resulting nucleotide sequences were registered with NCBI using text-based format for representing either nucleotide sequences (FASTA).

### 5.9. Statistical Analysis

The experiment was conducted in quintuplicate. The amounts of AFs were averaged and shown as mean ± standard error of the mean. With the support of statistical software (version 9.4; SAS, Institute Inc., Cary, NC, USA) a one-way analysis of variance (one-way ANOVA) and 95% confidence interval were performed. Tukey’s honest significant difference test (Tukey’s HSD) was used to estimate the significance of differences among treatment means. 

Efficacy was estimated as the percentage ratio of AF concentration achieved by aflatoxigenic strains in the presence of the non-aflatoxigenic CS1 strain compared to the aflatoxigenic strains inoculated separately on the same substrate, according to the following formula: E = ((AF^+^ - AF^−^)/AF^+^) × 100, where E is the efficacy, AF^+^ is the concentration of AFs produced by the aflatoxigenic strain and AF^−^ is the concentration of AFs produced when coinoculated with CS1. Additionally, a general linear-model analysis was performed to evaluate the efficacy (E) of isolate CS1 to reduce AF concentration as a combined effect of the substrates (S: WP, IG, CS, and CM) and the treatment (T: interaction of each toxigenic strain with CS1) nested with each substrate, in the fitted general linear model: E = S T(S). In all statistical analyses, a *p*-value < 0.05 was accepted as a significant difference.

## Figures and Tables

**Figure 1 toxins-14-00437-f001:**
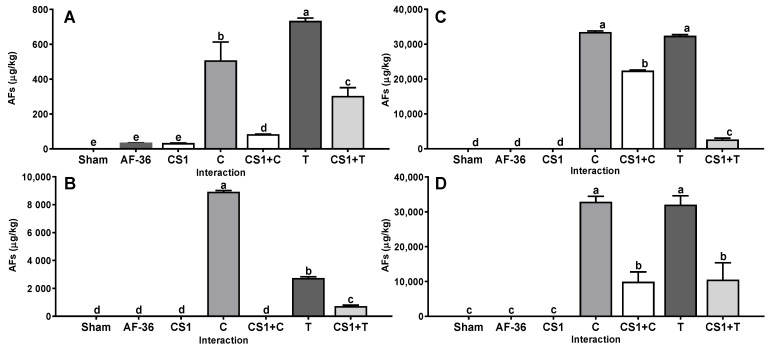
Interaction between *A. flavus* strain isolated from corn silage 1 (CS1) and the aflatoxigenic strains Cuautitlán (C) and Tamaulipas (T). (**A**) Whole plant; (**B**) corn silage; (**C**) immature grain; (**D**) culture media. A negative control (sham without inoculum) and the non-aflatoxigenic strain AF-36 are included; ^a–e^ Different literals indicate significant statistical differences (*p* < 0.05) among means of aflatoxins (AFs) concentration per strain.

**Figure 2 toxins-14-00437-f002:**
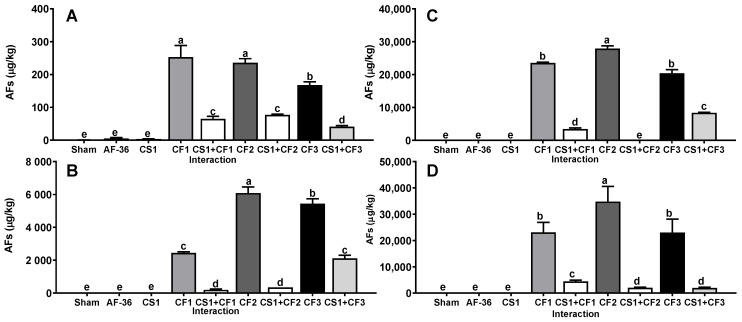
Interaction between *A. flavus* strain from corn silage 1 (CS1) and the aflatoxigenic strains isolated from concentrate feed (CF1, CF2, CF3). (**A**) Whole plant; (**B**) corn silage; (**C**) immature grain; (**D**) culture media. A negative control (sham without inoculum) and the non-aflatoxigenic strain AF-36 are included; ^a–e^ Different literals indicate significant statistical differences (*p* < 0.05) among means of aflatoxins (AFs) concentration per strain.

**Figure 3 toxins-14-00437-f003:**
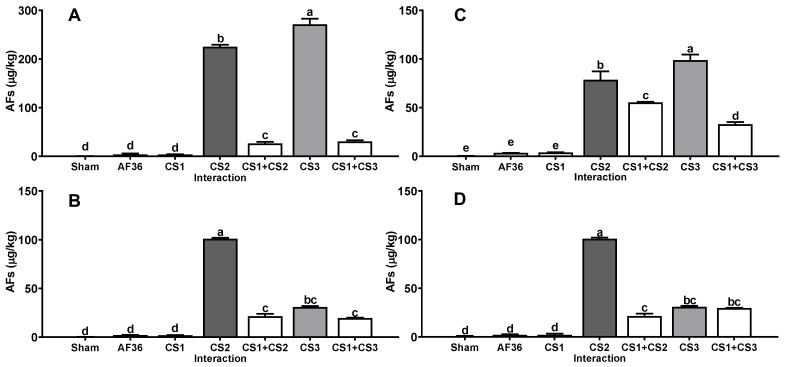
Interaction between *A. flavus* strain from corn silage 1 (CS1) and the aflatoxigenic strains isolated from corn silage (CS1, CS2). (**A**) Whole plant; (**B**) corn silage; (**C**) immature grain; (**D**) culture media. A negative control (sham without inoculum) and the non-aflatoxigenic strain AF-36 are included; ^a–e^ Different literals indicate significant statistical differences (*p* < 0.05) among means of aflatoxins (AFs) concentration per strain.

**Figure 4 toxins-14-00437-f004:**
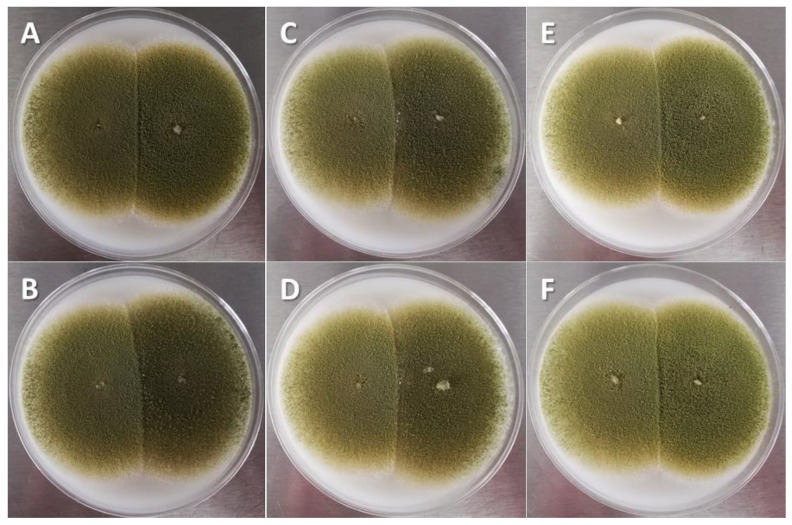
Interaction between nontoxigenic *A. flavus* strain isolated from corn silage 1 (CS1) and aflatoxigenic *A. flavus* strains in culture medium (5 days, coconut agar medium). Aflatoxigenic strains on the right of each image: (**A**) Cuautitlán; (**B**) Tamaulipas; (**C**) concentrate feed 1; (**D**) concentrate feed 3; (**E**) corn silage 2; (**F**) corn silage 3.

**Figure 5 toxins-14-00437-f005:**
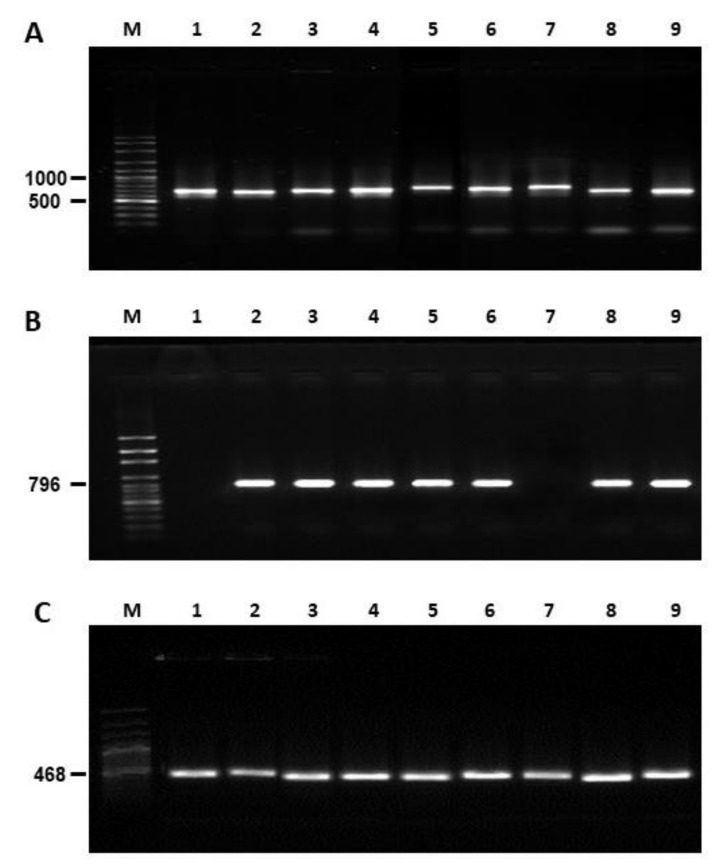
Gel electrophoretic analysis of PCR products using DNA obtained from *Aspergillus flavus* strains isolated from concentrate feed (CF), corn silage (CS) and control strains. (**A**) Internal transcribed spacer region; (**B**) calmodulin gene; (**C**) aflatoxins biosynthetic pathway regulator gene. Lanes: M: DNA molecular size markers (ladder in base pairs); 1: AF-36 (nontoxigenic control); 2: Cuautitlán (toxigenic control); 3: Tamaulipas (toxigenic control); 4: CF1; 5: CF2; 6: CF3; 7: CS1; 8: CS2; 9: CS3.

**Figure 6 toxins-14-00437-f006:**
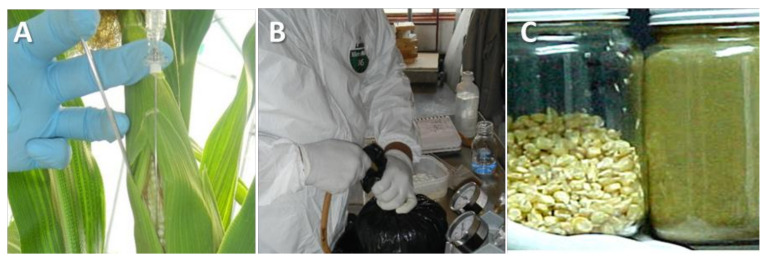
Design of interaction model between nontoxigenic *A. flavus* strain isolated from corn silage 1 (CS1) and toxigenic *A. flavus* strains on different substrates. (**A**) Whole plant (inoculation of the strains inside corn ear); (**B**) microsilage (application of vacuum on ground plant material inocu-lated with *A. flavus* strains); (**C**) immature grain (sham and Tamaulipas strains).

**Table 1 toxins-14-00437-t001:** Means ± CI 95% for efficacy (%) of corn silage isolate 1 (CS1) in reducing aflatoxin production.

CS 1 VS:	A. Whole Plant		B. Immature Grain		C. Corn Silage		D. Culture Media		General	
	Mean	CI 95%	Mean	CI 95%	Mean	CI 95%	Mean	CI 95%	Mean	CI 95%
All strains	76.0 ^A^	73.9–78.2	66.1 ^AB^	63.8–68.4	69.4 ^B^	65.1–73.7	62.5 ^AB^	57.4–67.6	68.5	66.9–70.1
C	79.1 ^ab^	73.5–84.8	33.0 ^d^	26.9–39.1	69.5 ^b^	68.4–70.5	70.7 ^a^	57.2–84.2	63.1 ^ab^	58.8–67.4
T	59.2 ^c^	53.5–64.9	91.8 ^ab^	85.7–97.9	65.6 ^b^	61.0–70.1	69.4 ^a^	55.9–82.9	71.5 ^ab^	67.2–75.8
CF1	73.8 ^b^	68.1–79.5	85.3 ^b^	79.3–91.4	91.8 ^a^	89.4–94.2	79.8 ^a^	66.2–93.3	82.7 ^a^	78.4–87.0
CF2	67.1 ^bc^	61.4–72.8	99.3 ^a^	93.2–100	94.3 ^a^	93.9–94.6	94.0 ^a^	80.5–100	88.7 ^a^	84.4–92.9
CF3	75.7 ^ab^	70.1–81.4	58.7 ^c^	52.6–64.8	61.2 ^b^	68.5–70.3	90.8 ^a^	77.3–100	71.6 ^ab^	67.3–75.9
CS2	88.4 ^a^	82.7–94.1	27.8 ^d^	21.7–33.9	70.4 ^b^	68.5–72.3	17.8 ^b^	4.3–31.3	51.1 ^b^	46.8–55.4
CS3	88.9 ^a^	83.2–94.6	66.8 ^c^	60.7–72.9	33.1 ^c^	28.2–37.9	15.3 ^b^	1.8–28.8	51.0 ^b^	46.7–55.3
R^2^ (%) *	71.7		97.7		97.3		87.9		90.6	
*p-value*	<0.01		<0.01		<0.01		<0.01		<0.01	

C = Cuautitlán; T = Tamaulipas; CF = concentrate feed; CS = corn silage. ^a–d^ Different lowercase letters indicate a significant statistical difference (*p* < 0.05) among treatments for each substrate (columns) or ^A,B^ lowercase letters in all strains (first row). * R square: coefficient of determination to evaluate the efficacy (E) of isolate CS1 to reduce AF concentration as the combined effect of substrate S (whole plant, immature grain, corn silage and culture media) and treatment T (interaction of CS1 with toxigenic strains) nested with S in the fitted general lineal model: E = S T(S).

**Table 2 toxins-14-00437-t002:** Design of treatments for the coinoculation of aflatoxigenic and non-aflatoxigenic strains of *Aspergillus flavus*.

No.	Isolate	ID	Morphotype	Aflatoxins Production	Interaction
1	Control ^1^	Sham		--	--
2	AF-36 (Negative control)	AF36	L	Negative	--
3	Cuautitlán (positive control)	C	S	Positive	CS1 + C
4	Tamaulipas (positive control)	T	S	Positive	CS1 + T
5	Concentrate Feed- 1	CF1	S	Positive	CS1 + CF1
6	Concentrate Feed-2	CF2	S	Positive	CS1 + CF2
7	Concentrate Feed-3	CF3	S	Positive	CS1 + CF3
8	Corn Silage-1	CS1	L	Negative	--
9	Corn Silage-2	CS2	S	Positive	CS1 + CS2
10	Corn Silage-3	CS3	S	Positive	CS1 + CS3

^1^ Control: application of the inoculum diluent without spores. Morphotype L = long sclerotium (>400 μm); Morphotype S = short sclerotium (<400 μm).

**Table 3 toxins-14-00437-t003:** Accession codes of nucleotide sequences at the National Center for Biotechnology Information^1^ for the internal transcribed spacer region (ITS) fragments, the AFs biosynthetic pathway regulator gene (aflR), and the calmodulin gene (CaM), which were amplified in experimental isolates of *Aspergillus flavus*, using the following forward (F) and reverse (R) primers: aflR-F aflR-R, ITS-1, ITS-4, cmdA7-F and cmdA8-R.

Isolate	ITS-1	ITS-4	aflR-F	aflR-R	cmdA7-F	cmdA8-R
C	ON351284	ON351498	MN987040.1 ***	AF441434.1 ***	(NS)	(NS)
T	ON351288	ON351503	CP051029.1 ***	KY769956.1 ***	(NS)	(NS)
CF1	HQ844707.1 ***	ON351496	XM_041285628.1 ***	L32577.1 ***	CP051084.1 ***	MK119700.1 ***
CF2	ON351282	ON351497	HQ844707.1 ***	MH511139.1 ***	CP051020.1 *	(NS)
CF3	ON351283	ON351501	MH752564.1 **	EF565462.1 ***	MN987032.1 ***	MK119699.1 ***
CS1	ON351285	ON351499	---	---	CP044620.1 **	CP051060.1 ***
CS2	ON351286	ON351500	KY769955.1 ***	AF441432.1 ***	CP044622.1 **	(NS)
CS3	ON351287	ON351502	MG720232.1 **	MH280087.1 **	CP051036.1 ***	MK119701.1 ***

^1^ NCBI: https://www.ncbi.nlm.nih.gov/nuccore/ (accessed on 12 May 2022) C = Cuautitlán; T = Tamaulipas; CF = concentrate feed; CS = corn silage *, **, *** Percent of identity with pre-existing nucleotide sequences: * ≥90, ** ≥95, *** ≥99; no asterisk indicates the accession codes for nucleotide sequences in this study. --- = without expression NS: No significant similarity found.

## Data Availability

Not applicable.

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
