# Peer review of "Assessment of the Potential of a Native Non-Aflatoxigenic Aspergillus flavus Isolate to Reduce Aflatoxin Contamination in Dairy Feed"

_toxins, 2022, doi:10.3390/toxins14070437_

Round 1

Reviewer 1 Report

Based on ref 19, 20, in this study the authors further explored the potential application of co-inoculation of a non-aflatoxigenic strain with different aflatoxigenic strains as an approach to reduce the production of toxic mycotoxins in feeds. The study design appears to be practical and logical with sufficient controls included. Presented data are of satisfactory quality, supporting the hypothesis. I would suggest removing the text from lines 83-86 from the introduction. The study has nothing to do with the safety issue of the non-aflatoxigenic strain.  

Author Response

Thank you for your comments.

The text relating to the safety of experimental isolate CS1 has been deleted (New lines: 110-112).

Reviewer 2 Report

     The study evaluated the efficacy of co-inoculation of a non-aflatoxigenic strain of A. flavus (CS1) isolated from corn silage in reducing aflatoxin formation by other stains of the same fungus, in various experimental conditions.

     The manuscript is written in a coherent manner.

      Co-inoculation of the CS1 strain resulted in moderate reduction of aflatoxins levels, so its use for the stated potential applications is debatable.

      Issues to resolve:

a) Lines 6-11, in the Abstract: The objective of the study is stated two times, in slightly different formulations. Please eliminate one of the two phrases or rephrase.   

b) In the Discussion section, please comment on the possibility of / challenges in standardising the CS1 strain for its stated potential uses.    

Author Response

Thank you for your comments.

The duplicate objective was eliminated (New lines: 6-8).

Text regarding the challenges and possibilities and challenges for establishing and utilizing the potential of the test isolate was included in the discussion section of the manuscript (New lines: 345-358).

Reviewer 3 Report

In this work the authors described the potential of a non-aflatoxigenic Aspergillus flavus against Aflatoxins. The manuscript requires some revisions to do:

-          The major problem is related to the lack of some key figures. From line 124 to line 130, phrase “The native strain of A. flavus CS1 was considered non-aflatoxigenic (Table 2) because  it did not produce AF nor was expression of the aflR gene observed; however, in all experimental isolates and positive control isolates (Cuautitlán and Tamaulipas) expression  of this regulatory gene of the AF biosynthetic pathway was observed. All isolates were  identified as A. flavus by analysis of gene expression of an ITS region and CaM (Table 3). The nucleotide sequences obtained showed a high percentage of homology (90-99 %) with  other nucleotide sequences registered for A. flavus in public databases”, the authors described the results they obtained from molecular analysis. Anyway, it is not enough only to insert a table ( Table 3) in the manuscript. The authors should insert in the manuscript some pictures related to the molecular analysis. In detail, in which way and what are the figures proving what the authors mentioned from line 124 to line 127 “The native strain of A. flavus CS1 was considered non-aflatoxigenic (Table 2) because  it did not produce AF nor was expression of the aflR gene observed; however, in all experimental isolates and positive control isolates (Cuautitlán and Tamaulipas) expression of this regulatory gene of the AF biosynthetic pathway was observed”? Do the authors have some figures related to PCR?

-          The authors should check the English grammar in the manuscript. There are some phrases which are not very clear. For example “The native strain of A. flavus CS1 was considered non-aflatoxigenic (Table 2) because  it did not produce AF nor was expression of the aflR gene observed” line124.

-          From line 54 to line 61, the authors  described there are several methods to prevent the contamination. Anyway, they can just mention in a short sentence the use of antioxidants as feed additive in reducing the negative effect due to aflatoxin exposure ( for example DOI: 10.3389/fvets.2021.822227) . In this way, the authors will mention all the strategies to prevent AF exposure and reduce their toxicity and make the manuscript more updated for the readers.

Other minor revisions to do:

-          Line 36 and Line 41, please replace AF with AFs. The authors should check in the entire manuscript when it is necessary.

-          Line 44, the authors should write in a better way this part “when ingested in food by humans or animals”

-          The phrase “The epoxides are highly reactive and thus, electrophilically attack and damage to subcellular structures and nucleic acids are produced” in lines  47-48 is not well structured and clear. Please, the authors should write this phrase in a better way.

-          The authors should add in the figure 1A and 1C the title of x axis. And the name of the experimental groups as done in Figure 1B and 1D. The authors should add in the figure 1C and 1D the name of y-axis.  The authors should do the same thing for the figure 2 and figure 3.

Author Response

Thank you for your comments.

A figure of the PCR products of the genes and regions of each isolate included in the study was added (New lines: 231-238).

A summary of strategies used to prevent or reduce aflatoxin exposure was included (New lines: 72-85).

The expression "AF" was replaced by "AFs" throughout the manuscript.

The text referring to line 44 was reformulated (New line: 42).

The grammar of the manuscript was rechecked. The text referring to line 124 was reformulated (New lines: 192-193).

The text concerning lines 47-48 was rephrased (New lines: 62-63).

All experimental group names and X- and Y-axis titles were inserted in Figures 1, 2 and 3.

Round 2

Reviewer 3 Report

the work is ready to be published